# In Vitro Efficacy of Essential Oils from *Melaleuca Alternifolia* and *Rosmarinus Officinalis*, Manuka Honey-based Gel, and Propolis as Antibacterial Agents Against Canine *Staphylococcus Pseudintermedius* Strains

**DOI:** 10.3390/antibiotics9060344

**Published:** 2020-06-19

**Authors:** Gabriele Meroni, Elena Cardin, Charlotte Rendina, Valentina Rafaela Herrera Millar, Joel Fernando Soares Filipe, Piera Anna Martino

**Affiliations:** 1Department of Biomedical Sciences for Health, University of Milan, Via Mangiagalli 31, Milan 20133, Italy; 2Department of Veterinary Medicine, University of Milan, Via dell’Università 6, Lodi 26900, Italy; elenacardin.vet@gmail.com (E.C.); charlotterendina@icloud.com (C.R.); valentina.herrera@unimi.it (V.R.H.M.); joel.soares@unimi.it (J.F.S.F.); piera.martino@unimi.it (P.A.M.)

**Keywords:** antibiotic resistance, *melaleuca alternifolia*, MRSP, multidrug resistance, virulence factors

## Abstract

Essential oils (EOs) and honeybee products (e.g., honey and propolis) are natural mixtures of different volatile compounds that are frequently used in traditional medicine and for pathogen eradication. The aim of this study was to evaluate the antibacterial properties of tea tree (*Melaleuca alternifolia*) EO (TTEO), *Rosmarinus officinalis* EO (ROEO), manuka-based gel, and propolis against 23 strains of *Staphylococcus pseudintermedius* (SP) isolated from canine pyoderma. Antimicrobial resistance screening was assessed using a panel of nine antimicrobial agents coupled with a PCR approach. An aromatogram was done for both EOs, using the disk diffusion method. The minimum inhibitory concentration (MIC) was determined for all the compounds. Among the 23 SP strains, 14 (60.9%) were multidrug-resistant (MDR), 11 strains (47.8%) were methicillin-resistant (MRSP), and 9 (39.1%) were non-MDR. The mean diameter of the inhibition zone for *Melaleuca* and *Rosmarinus* were 24.5 ± 8.8 mm and 15.2 ± 8.9 mm, respectively, resulting as statistically different (*p* = 0.0006). MIC values of TTEO and ROEO were similar (7.6 ± 3.2% and 8.9 ± 2.1%, respectively) and no statistical significances were found. Honeybee products showed lower MIC compared to those of EOs, 0.22 ± 0.1% for Manuka and 0.8 ± 0.5% for propolis. These findings reveal a significant antibacterial effect for all the tested products.

## 1. Introduction

*Staphylococcus pseudintermedius* (SP) is a Gram-positive bacterium that can act both as a skin commensal or a pathogen and is nowadays considered one of the main causative agents of canine pyoderma [1,2]. It is also able to cause ear, wound, and post-surgical infections in domestic animals, particularly in dogs [3,4,5]. From the first documented identification of this pathogen [6], SP was considered to be exclusive to animals until [7] described the first case of human infection. As it belongs to the *Staphylococcus* genus, SP can easily exchange genomic material [8,9], and in this context, the spread of antibiotic resistance genes (ARG) is a huge challenge for both human and animal health [10,11,12,13]. The evolution of methicillin-resistant SP strains (MRSP, *mecA*-positive) rapidly emerged as a significant health problem in veterinary medicine because of the molecular mechanism responsible for β lactam-resistance (SCC*mec* element) that is transferable between different species of staphylococci [14,15,16]. Furthermore, MRSP strains are often resistant to more than three classes of antibiotics (multidrug-resistant (MDR)). This antibiotic resistance applies to all classes of antimicrobials, and is mainly mediated by resistance genes carried by mobile genetic elements (MGE) [13,17]. Some resistances (e.g., to fluoroquinolone) are linked to single nucleotide polymorphism (SNP) [18]. Nowadays, the zoonotic potential of SP is under constant investigation, and different authors have studied its virulence factors, including its ability to form biofilms [3,13,19]. 

For centuries, humans have used natural products to treat common diseases, as a natural medication, or as a supplement in diets [20,21,22,23]. Essential oils [EOs) represent the principal components of aromatherapy, and up to 17,000 different species of plants from 60 different families (e.g., *Lamiaceae*, *Rutaceae*, *Myrtaceae*, *Zingiberaceae*, and *Asteraceae*) produce them [24,25]. EOs are defined as volatile mixtures of organic compounds derived from steam distillation of plants (flowers, seeds, leaves, and roots) [26,27,28,29]. The antimicrobial properties of EOs depend on terpenes, terpenoids, and aromatic and aliphatic groups, which are the main constituents of oils after the process of distillation. 

Different studies investigated the antimicrobial properties of EOs and found that Gram-positive bacteria are more susceptible than Gram-negative bacteria [25]. Tea tree EO (TTEO) is derived from the distillation of leaves and twigs of *Melaleuca alternifolia*, which contains the following different volatile compounds: terpine-4-ol (≥30%), teripene (about 20%), α-terpinene (about 8%), ρ-cymene (about 8%), α-pinene (about 3%), terpinolene (about 3%), and 1,8-cineol (≤15%) [30]. In terms of medical and aromatic value, rosemary (*Rosmarinus officinalis* L.) is of considerable importance. Major volatile components of rosemary essential oil (ROEO) include alpha-pinene (24%), camphen (9%), camphor (11%), verbenon (15%), p-cymene (8%), and 3-octanone (6%) [24]. 

Honey is a natural product derived from honey bees, with high nutritional values for human health, in particular antioxidant, bacteriostatic, anti-inflammatory, and antimicrobial properties, as well as wound and sunburn healing effects [28]. Manuka honey is a monofloral honey (*Manuka* tree), with unique properties, mainly antimicrobial [31]. Its antimicrobial properties regard the ability to inhibit bacterial growth of both Gram-positive and Gram-negative bacteria, including *Staphylococcus aureus*, *Streptococcus pyogenes*, *Pseudomonas aeruginosa*, and *Escherichia coli*, making Manuka honey a broad-spectrum agent [32,33]. Manuka honey is classified using a system known as the Unique Manuka Factor (UMF), which indicates the equivalent concentration of phenol (%, w/v) needed to obtain the same antibacterial activity as honey [31]. The UMF is directly proportional to the honey’s antibacterial properties; different researchers have found that bacteriostatic ability shown by different grades of manuka honey is evident at concentrations of 10% and 20% [34,35]. Propolis (also known as bee glue), is a sticky, resinous substance produced by honey bees from different plants [36]. This product consists of 45–55% plant resin, 25–35% wax, 5–10% essential and aromatic oils, 5% pollen, and 5% of other natural products [37]. 

As a primary outcome, this study aimed to evaluate the antibacterial effects of the following natural products: tea tree and rosemary essential oils, manuka honey-gel, and propolis against strains of *Staphylococcus pseudintermedius* isolated from canine pyoderma. Secondary outcomes are the evaluation of the antibiotic resistance profile and virulence potential to better understand the possible zoonotic potential of the isolates.

## 2. Results

### 2.1. Antibacterial Activity of Natural Products

#### 2.1.1. Aromatogram

The antibacterial abilities of EOs used are summarized in Figure 1. All datasets had a Gaussian distribution as resulted from four different tests: Anderson–Darling test, D’Agostino and Pearson test, Shapiro–Wilk test, and Kolmogorov–Smirnov test. Inhibition diameters of TTEO (24.5 ± 8.8 mm) resulted in statistical significance (*p* = 0.0006) from those of ROEO (15.2 ± 8.9 mm) (Figure 1A). The same statistical trend was maintained between MDR (*N* = 14) and non-MDR (*N* = 9) strains (Figure 1B). In detail, the statistical significances between TTEO and ROEO in the MDR group and non-MDR group were *p* = 0.035 and *p* = 0.0056, respectively. One MDR strain did not show a measurable diameter for each of the EOs; the same finding was observed for one non-MDR strain that was considered resistant to rosemary oil.

#### 2.1.2. MIC of Natural Products

The minimum inhibitory concentrations of the EOs (%; v/v), manuka-based gel, and propolis are shown in Figure 2. Each of the datasets resulted in non-normal distributions, and a non-parametric test (Mann–Whitney test) was used to check statistical significances. Statistical analysis for EOs did not find any difference between all the strains (Figure 2A). TTEO had a mean minimum inhibitory concentration (MIC) value of 7.6 ± 3.2%, while ROEO had a value of 8.9 ± 2.1%. Furthermore, the same statistical trend was found among MDR and non-MDR strains (Figure 2B).

Manuka-based gel (C) showed lower MIC (v/v) values (0.22 ± 0.1%) compared to those of propolis (0.8 ± 0.5%), resulting in statistical significance (*p* < 0.0001). Among MDR strains (D), manuka was statistically different from propolis (*p*-value < 0.0001).

Figure 3 shows the logarithmic transformation of MIC values to allow better visualization of the results. Multiple comparisons with Kruskal–Wallis test were performed.

Each of the EOs were statistically different from both manuka and propolis (*p*-value < 0.0001), displaying higher MIC values. Moreover, manuka-based gel was the natural product with the lowest MIC, resulting in a statistical significance compared to propolis (*p*-value < 0.0001).

### 2.2. Resistance to Antimicrobial Agents

Twelve (52.2%) of the isolates were resistant to amoxicillin and amoxicillin + clavulanic acid, 11 (47.8%) to clindamycin, 14 (60.9%) to cefovecin, 8 (34.8%) to tetracyclines, 13 (56.5%) to enrofloxacin, and 10 (43.5%) to marbofloxacin.

Among the 23 SP strains, the majority (14/23, 60.9%) were resistant to more than 3 pharmacological categories and were classified as MDR. In particular, 11/14 (78.6%) were resistant to oxacillin and harbored the *mecA* gene [coding the penicillin binding protein 2a] (MRSP strains); the remaining 3/14 (21.4%) were considered as methicillin-susceptible SP (MSSP). Nine strains out of 23 (39.1%) were susceptible to antibiotics (not MDR); 6/9 (66.7%) to all the molecules, whereas 3/9 (33.3%) showed a multifaceted susceptibility pattern. The amplification of ARG confirmed such profiles (Table 1).

### 2.3. Genotyping

The genetic characteristics of the 23 SP strains, together with other information (antibiotic resistance profile and dissemination of virulence genes), are reported in Table 1. The MultiLocus Sequence Type (MLST) showed eight different sequence types (STs): ST71, ST106, ST44, ST100, ST28, ST127, ST108, and ST26. The most frequently found ST was ST71 in 7/23 (30.4%) strains, followed by ST 106 in 4/23 (17.4%), and ST44 in 6/23 (26.1%). The MDR strains belong to STs 71, 106, and 26, while the strains susceptible to all the antibiotic molecules were classified as ST44. The remaining isolates included one isolate belonging to ST100, one to ST28, two to ST108, one to ST127, and one to ST26. 

Among the 23 strains, we applied the SCC*mec* typing to 11 isolates (47.8%) because of methicillin-resistant strains. Two SCC*mec* types were found: II-III and IV. The most prevalent was SCC*mec* type II-III, identified in 7/11 (63.6%) of the strains, while the remaining 4/11 (36.4%) were classified as SCC*mec* type IV.

### 2.4. Virulence Factors

Summary of PCR-analysis of the 23 SP strains is shown in Table 1. Bicomponent leukocidin (*lukS/F-I*) was found in 10/23 (43.5%) strains; all of them were MDR. Furthermore, 14/23 (60.9%) strains were *sec_canine_*-positive and 18/23 (78.3%) were *se-int*-positive, and species-specific exfoliative toxin (*siet*) was amplified in all the isolates. Biofilm-related gene *icaA* [intercellular adhesion gene A] was found in 16/23 (69.5%) isolates, while *icaD* [intercellular adhesion gene D] was found in all the strains. Table 2 reports the detailed dissemination of predicted virulence factors among MDR and non-MDR strains.

## 3. Discussion

Antimicrobial resistance remains one of the most crucial challenges to be solved in both human and veterinary medicine. The use of natural products to treat common infectious diseases is one of the oldest practices in human history [25,29]. This study investigated the antimicrobial properties of two EOs and two honeybee products against *S. pseudintermedius* strains isolated from canine pyoderma. Clonal relatedness confirmed the findings of previous studies in which MRSP ST71 is considered the most prevalent in Europe, even though a novel MRSP clone (ST106) is rapidly spreading in North Europe (Norway) [8,13,38,39,40]. The identification of ST 44 as one of the main STs susceptible to the majority of antibiotic molecules, with a prevalence of 26.1%, was in accordance with the study of Gharsa (2013); this author analyzed 55 SP strains—all of them were MSSP, and 27% were susceptible to all the antibiotics tested [40]. MRSP represents 47.8% of our strains; the pattern of antibiotic resistance was similar to those reported by [17], with multiple resistances against β-lactams, tetracyclines, and fluoroquinolones. Resistance to methicillin was confirmed by mecA amplification, and SCC*mec* typing showed two main SCC*mec* types, both found in the literature: SCC*mec* II-III and IV. The SP strains that harbor these two chromosomal cassettes are thought to be more susceptible to zoonotic transmission, as found in the study by [11].

Essential oils and honeybee products are widely used in veterinary medicine as arthropod repellents, for topical administration (e.g., shampoo), and as antimicrobials and antifungals [41,42,43,44]. It was already known that the antimicrobial abilities of EOs are strictly dependent on the environmental characteristics that affect plant development [25,29,45]. On the basis of the results from the aromatogram, tea tree EO was able to generate larger inhibition zones compared to those of rosemary. One explanation for this result could be the unique and peculiar chemical composition of tea tree oil that gives it its broad-spectrum antibacterial abilities [25,30,46,47]. Tea tree EO showed in vitro antibacterial properties against *Porphyromonas gingivalis*, *Porphyromonas endodontalis*, *S. aureus*, *E. coli*, *Streptococcus mutans*, and *Listeria monocytogenes* [25,30,46,47]. The ability to inhibit bacterial growth was studied by the determination of MICs for both EOs. However, no statistical significances were found among the two oils. In the literature, the scientific evidence about the topical use of tea tree against SP is poor, but rosemary is currently used. Our results seem to be partially concordant with the available literature [48]. Rosemary essential oil showed antimicrobial properties against *L. monocytogenes*, *E. coli*, and *Salmonella enterica* Serovar Enteritidis [45,49]. Among the two honeybee products tested (manuka-based gel and propolis), our results demonstrated a high rate of concordance with the literature [30,34,50]. Manuka-based gel was found to have the lowest MIC, making this product one of the most promising natural antimicrobials. The mean MIC value was 0.22 ± 0.1%, lower than those reported by [50], who used essential oil of *Leptospermum scoparium* (manuka) against MRSP and MSSP and found MIC values of 2% and 8%, respectively. In another study on *S. aureus*, the authors of [34] found an evident inhibitory effect of honey at concentrations of 10% and 20%. It has to be noted that our tested product was not pure manuka honey, but a commercial preparation for topical use to promote wound healing. The mean MIC value for propolis was slightly higher than that of manuka, at 0.8 ± 0.5%. In the literature, it is reported that propolis shows antibacterial effects against both Gram-positive and Gram-negative bacteria, including methicillin-resistant *Staphylococcus aureus* (MRSA), but also that it is effective against fungi and yeast [51,52,53].

Independently of their genetic background or resistance profile, we detected a variety of virulence factors among our SP collection. One of the most interesting findings is that MDR strains did not necessarily harbor more virulence factors. Specific enterotoxin *se-int* and biofilm-producing gene *icaD* were found in all non-MDR strains, with significantly less prevalence in the MDR group. These results, taken together, should suggest paying attention, not only to antibiotic-resistance but also to the virulence potential of the strains that could easily become zoonotic agents.

## 4. Conclusions

Essential oils and honeybee-derived products should be considered a potential alternative to conventional antibiotics; in some cases, they represent the raw materials for antibiotics production. This in vitro study demonstrated the strong antibacterial properties of these products against *S. pseudintermedius*. However, more precise in vivo studies should be designed to fully characterize their properties and understand the potentially dangerous effect that some essential oils could have after systemic administration.

## 5. Materials and Methods

### 5.1. Bacterial Isolation and Idenfication

SP strains from 23 different dogs, with clinical symptoms referable to canine pyoderma, were collected from clinical samples (cutaneous swabs) submitted for routine microbiological diagnosis to the Microbiology and Mycology laboratory of the Department of Veterinary Medicine, University of Milan. After collection, swabs underwent a standardized laboratory workflow for bacterial identification. The samples were plated on Tryptic Soy Agar (TSA; ThermoFisher, Milan, Italy) + 5% of defibrinated sheep blood (ThermoFisher, Milan, Italy) and were incubated aerobically at 37 °C for 24 h. Suspected staphylococcal colonies were identified by conventional methods (Gram staining, catalase test), and isolated colonies were subcultured on Mannitol Salt Agar (MSA; ThermoFisher, Milan, Italy) and incubated aerobically at 37 °C for 24 h. The phenotypic identification of SP was also confirmed at a genetic level by restriction fragment length polymorphism analysis (RFLP-PCR), as described previously [54], and by the amplification of the thermonuclease gene (*nuc*) using specific primer pairs, as found in the literature [55]. Pure cultures were stored in 25% glycerol (Carlo Erba, Milan, Italy) at −20 °C.

### 5.2. DNA Extraction

Pure cultures in 25% glycerol were thawed at room temperature and grown in brain heart infusion agar (BHI; Microbiol, Cagliari, Italy) at 37 °C for 24 h. Three to four isolated colonies, from each culture, were removed, and DNA was extracted using the boiling method described in the literature [56]. The purity and concentration of nucleic acid was checked using Qubit (ThermoFisher, Milan, Italy).

### 5.3. Molecular Typing and Clonal Relatedness

Two standard molecular typing techniques were used to analyze the genetic correlation of SP strains: MultiLocus Sequence Typing (MLST) and SCC*mec* Typing. MLST was performed using the 7-gene MLST workflow found in the *Staphylococcus pseudintermedius* MLST database (https://pubmlst.org/spseudintermedius/). The amplification of 7 specific genes (*ack* [acetate kinase], *cpn60* [chaperonin 60], *fdh* [formate dehydrogenase], *pta* [phosphoacetyltransferase,], *purA* [adenylosuccinate synthetase], *sar* [sodium sulfate symporter], and *tuf* [elongation factor]) was carried out as described by [57]. Sequences were aligned using the National Center for Biotechnology (NCBI) nucleotide database to obtain the number of alleles. Sequence types (STs) were attributed as found in the literature [58], and using the online tool (https://pubmlst.org/spseudintermedius/).

A specific set of multiplex PCRs was used to assign SCC*mec* type I-IV among MRSP strains only; the amplification conditions, as well as the primer pairs used, were the same as those reported in the literature [59,60].

### 5.4. Antimicrobial Susceptibility Testing

#### 5.4.1. Kirby-Bauer Disk Diffusion and Detection of Antimicrobial Resistance Genes (ARG)

Susceptibility to 9 antimicrobial agents was performed using the Kirby-Bauer disk diffusion method, according to the Clinical and Laboratory Standard Institute guidelines [61]. Antimicrobials tested were amoxicillin + clavulanic acid (AMC; 20 + 10 µg), amoxicillin (AML; 30 µg), oxacillin (OX; 1 µg), cephalexin (CL; 30 µg), cefovecin (CVN; 30 µg), clindamycin (DA; 10 µg), enrofloxacin (ENR; 5 µg), marbofloxacin (MAR; 5 µg), and tetracycline (TE; 30 µg).

The presence of the following ARGs (*mecA* [methicillin resistance gene A], *blaZ* [beta lactamase resistance gene Z], *tetM* [tetracycline resistance gene M], *tetK* [tetraycline resistance gene K], *aacA-aphD* [aminoglycosides resistance]) was studied using two multiplex PCRs with primer pairs and amplification conditions found in the literature [62,63].

### 5.5. Aromatogram of Tea Tree and Rosmarinus Officinalis

Essential oils of tea tree (TTEO) and *Rosmarinus officinalis* (ROEO) were purchased in a local pharmacy in Milan. To assess the antimicrobial activity of these EOs, we used a modification of the classical disk diffusion method (aromatogram), following a protocol by [64]. Five microliters of each essential oil (TTEO and ROEO) were impregnated into sterile paper disks (6 mm diameter, ThermoFisher, Milan, Italy) and placed onto Mueller–Hinton agar plates (ThermoFisher, Milan, Italy) previously inoculated with 10^8^ (Colony Forming Unit per milliliter) CFU/mL suspension of each SP strain. The plates were then incubated aerobically at 37 °C for 24 h. After incubation, the antibacterial activity was evaluated by measuring the diameter of inhibitory zones in millimeters and expressed as mean ± SD. Four sterile paper disks were used for each EOs on the same agar plates, and two independent plates for each bacterial strain were used to evaluate the effects of a single EO.

### 5.6. Minimum Inhibitory Concentration (MIC)

#### 5.6.1. Tea Tree and Rosmarinus Officinalis

The minimum inhibitory concentration (MIC) for TTEO and ROEO was determined by following the microdilution method according to the Clinical and Laboratory Standard Institute guidelines [61]. As found in the literature [48], a 1% solution of Tween 80 (Merck, Milan, Italy) in Mueller–Hinton broth (ThermoFisher, Milan, Italy) was used to stabilize EOs; 100 µL was dispersed in the first column of a 96-well plate, and twofold serial dilutions (range from 10% to 0.019% v/v) were then created with a multichannel pipette. Proper suspensions of SP strains (equivalent to 0.5 McFarland standard) were prepared in sterile saline solution (0.9%) and, after dilution to 10^6^ UFC/mL, were transferred to the microdilution plate. The test was performed in duplicate for each strain on the same plate. After aerobic incubation at 37 °C for 24 h, we found MIC to be the lowest concentration able to inhibit visible bacterial growth. To facilitate this interpretation, we loaded 10 µL of 0.01% aseptic resazurin solution (viability marker) in each well (one of the two replicates) and further incubated them aerobically at 37 °C for 2–4 h, as done in the literature [65]. The identification of metabolically active bacteria was made, focusing on the change in color of the viability marker, with purple-pink wells indicating viable bacteria, and blue ones indicating metabolic stasis.

#### 5.6.2. Manuka Honey-Based Gel and Propolis

##### Manuka Honey-Based Gel

A commercially available manuka honey-based gel (Medihoney, Comvita) was used to determine the MIC value using the microdilution assay. In preliminary experiments (data not shown) focused on the handling of this viscous product, we concluded that a pre-warming step at 56 °C was necessary to avoid an incomplete dissolution of the manuka honey. A 40 mg/mL (1%, w/v) stock solution of manuka honey gel was obtained by dissolving this product in sterile Mueller–Hinton broth (ThermoFisher, Milan, Italy). After the preparation of bacterial suspensions and loading of the microtiter plate, we found the final concentration of the tested product in the first well to be 10 mg/mL (1%), and from the second until the tenth, we performed a 1:2-fold dilution (range from 1% to 0.0019%). The test was performed in duplicate for each strain on the same plate to assess the viability assay further. 

##### Propolis

A natural preparation of propolis in ethanol was purchased from a local pharmacy in Milan, which was used to study the antibacterial properties of this honeybee product. The concentration of ethanol was thought to be about 96%, on the basis of other food supplements that use ethanol as a solvent. To rule out the possibility that the bactericidal effect of propolis was due to the ethanol in which it was dissolved and not to the natural product itself, we performed a preliminary set of experiments (data not shown) to determine the MIC for 96% ethanol. Similarly, as for the honey-based gel, the propolis also underwent a preliminary pre-warming step at 56 °C immediately before use to allow its complete solubilization. The MIC was calculated according to the Clinical and Laboratory Standards Institute (CLSI) guidelines [42]. The final concentration of propolis in the first well was about 25%, with a range from 25% to 0.048%. The test was performed in duplicate for each of the strains on the same plate. Incubation was performed aerobically at 37 °C for 24 h. Due to the natural brownish color of the product, which made the determination of the MIC difficult, we performed the viability assay, as previously described.

### 5.7. Detection of Virulence Factors

All SP strains were tested by PCR for the presence of the enterotoxins (*sec_canine_*, *se-int*), exfoliative toxin (*siet*), bicomponent leukocidin (*lukS/F-I*), and biofilm forming genes belonging to ica locus (*icaA* and *icaD*); primers pairs, as well as amplification conditions, were the same as those reported in the literature [66,67,68,69,70].

### 5.8. Statistical Analysis

Statistical analysis and graphic representation were performed using GraphPad Prism (version 8.0.1 for Windows, GraphPad Software, San Diego, CA, USA, www.graphpad.com). Anderson–Darling, D’Agostino and Pearson, Shapiro–Wilk, and Kolmogorov–Smirnov tests were used to assess the normality of data. One-way ANOVA and multiple comparison tests of the data were performed to verify statistical significances between essential oils and honeybee products.

## Figures and Tables

**Figure 1 antibiotics-09-00344-f001:**
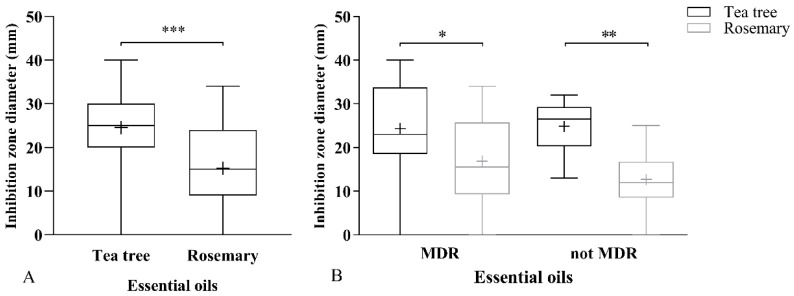
Aromatogram of tea tree and rosemary essential oils. Data are expressed using box and whisker plot, min to max values are presented by vertical lines, while median and mean are within the plot as horizontal line and “+”, respectively. An unpaired *t*-test was used to determine (**A**) statistical significances among the 23 *Staphylococcus pseudintermedius* (SP) strains; (**B**) stratification of the results over multidrug-resistant (MDR) (*N* = 14) and non-MDR (*N* = 9) strains. (* *p*-value between 0.05 and 0.01; ** *p*-value < 0.01, *** *p*-value < 0.001).

**Figure 2 antibiotics-09-00344-f002:**
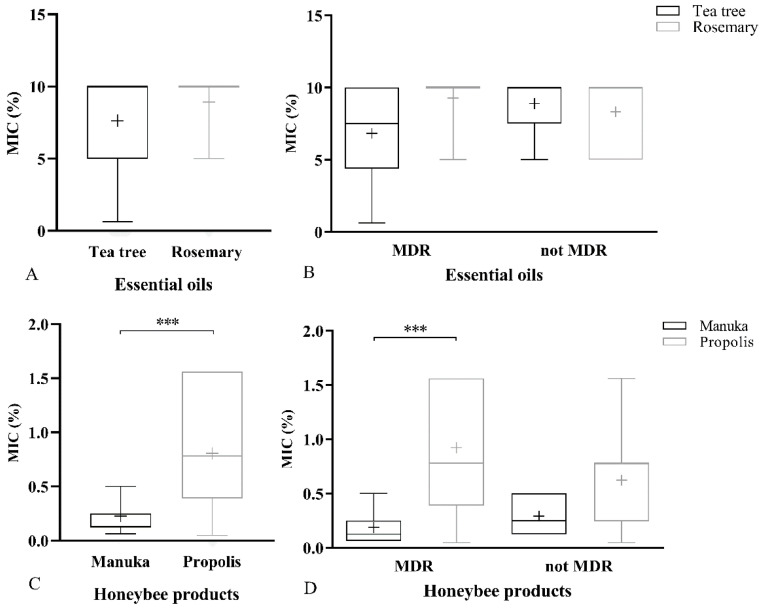
Minimum inhibitory concentration (MIC) values for all the natural products used. Box-plot was used to visualize the dataset, min to max are expressed by vertical lines, while median and mean are represented by horizontal and “+”, respectively; MIC are expressed as percentage volume per volume. *t*-test and one-way ANOVA followed by multiple comparison found no statistical significance between the two essential oils (EOs) (**A**) and among MDR and non-MDR strains (**B**). However, the same statistical tests found significative differences for honeybee products with lower MIC compared to those of EOs and a greater antibacterial ability. In particular, manuka-based gel (**C**) was statistically different compared to propolis even among MDR strains (**D**). (*** *p*-value < 0.001).

**Figure 3 antibiotics-09-00344-f003:**
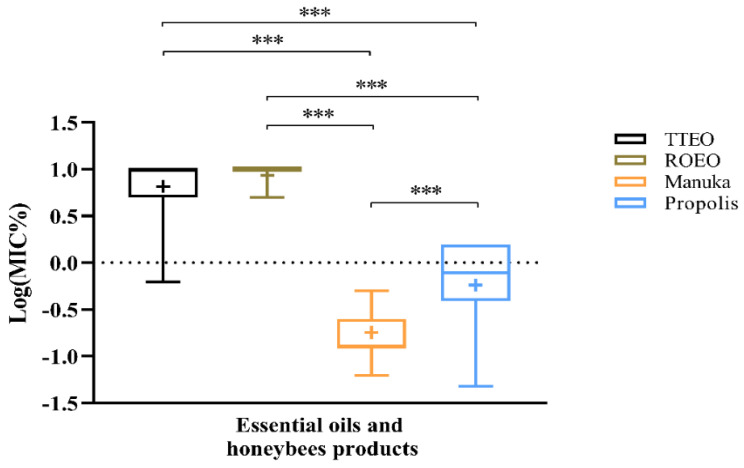
One-way ANOVA and multiple comparison test between MIC for different natural products. The box and whisker plot refers to the logarithmic transformation of raw data to improve the correlation of data with different magnitude. Data are presented as min to max, the horizontal line (within each plot) refers to median, while “+” indicates mean. (*** *p*-value < 0.001).

**Table 1 antibiotics-09-00344-t001:** Phenotypic and genotypic features of SP isolates.

No. of Isolates	Antimicrobial Resistance Profile	Resistance Genes Detected	Enterotoxin Genes Detected ^a^	Other Toxin Genes Detected ^a^	Biofilm Genes Detected ^a^	MLST ^a^	SCC*mec* Type ^a^
6	Susceptible	-	*se-int; sec^2^*	*siet*	*icaA; icaD*	ST 44	-
5	OX-AMC-AML-CL-ENR-MAR	*mecA; blaZ; tetM;aacA-aphD*	*se-int; sec*	*siet; lukS/F-I*	*icaA^2^; icaD^5^*	ST 106^2^; ST 71^3^	IV^1^; II-III^4^
4	OX-AMC-AML-CL-CVN-TE-ENR-MAR	*mecA; blaZ; tetM;tetK; aacA-aphD*	*se-int^3^; sec*	*siet; lukS/F-I^3^*	*icaA; icaD*	ST 106^2^; ST 71^2^	IV^1^; II-III^3^
2	TE	*tetM*	*se-int; sec^1^*	*siet*	*icaA; icaD*	ST 108	-
2	TE-ENR-MAR-CVN	*tetK*	*sec^1^*	*siet*	*icaA^1^; icaD*	ST 127; ST 100	-
1	CVN	*blaZ*	*se-int*	*siet*	*icaA; icaD*	ST 28	-
1	OX-AMC-AML-CL-CVN-TE-ENR	*mecA; blaZ; tetM*	*sec*	*siet; lukS/F-I*	*icaA; icaD*	ST 71	-
1	AMC-AML-CVN-ENR	*blaZ*	*se-int*	*siet*	*icaD*	ST 71	IV
1	OX-AMC-AML-CVN- TE-ENR-MAR	*mecA; blaZ; tetM*		*siet*	*icaA; icaD*	ST 26	IV

For antibiotic abbreviations, refer to Section 5.4.1. ^a^ Superscript indicates the number of strains with a specific virulence gene or genetic characteristic. The lack of this indication means that all the strains have this particular finding.

**Table 2 antibiotics-09-00344-t002:** Dissemination of virulence genes among MDR and non-MDR strains.

Virulence Genes	MDR (*N* = 14)	Non-MDR (*N* = 9)	*p*
*lukS/F-I*	10/14; (71.4%)	0/9; (0%)	-
*sec_canine_*	11/14; (78.6%)	3/9; (33.3%)	0.03
*se-int*	9/14; (64.3%)	9/9; (100%)	0.0427
*siet*	14/14; (100%)	9/9; (100%)	-
*icaA*	7/14; (42.8%)	9/9; (100%)	0.011
*icaD*	14/14; (100%)	9/9; (100%)	-

Yates’s chi-squared test was used to compare proportion of virulence determinants between MDR (*N* = 14) and non-MDR (*N* = 9) strains.

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
