# Peer review of "In Vitro Efficacy of Essential Oils from Melaleuca Alternifolia and Rosmarinus Officinalis, Manuka Honey-based Gel, and Propolis as Antibacterial Agents Against Canine Staphylococcus Pseudintermedius Strains"

_antibiotics, 2020, doi:10.3390/antibiotics9060344_

Round 1

Reviewer 1 Report

The topic of the research seem to be very important and crucial because of the need of searching for new antimicrobial agents.

The authors evaluated the antibacterial properties of essential oils from Tea tree (Melaleuca alternifolia) , Rosmarinus officinalis (ROEO), Manuka based gel and propolis against 23 strains of Staphylococcus pseudintermedius (SP) isolated from canine pyoderma. Moreover an aromatogram of EOs, using the disk diffusion method was done. The virulence potential of the strains was determined.

I recommend to publish the research but after some revisions.

First of all the form of presentation of antimicrobial data results, i.e. MICs data and aromatograms, in Figures 2 and 1, are not readable. Could you please to summarize the MICs values and Inhibition zone diameters in form of Table simply.

There is a lack of Conclusions, as a separate paragraph, in the manuscript. Could you complete it?

I have found in literature some essential positions (1-6 below) that are strongly connected with the studied topic, please include them into Introduction part or Discussion part.

  1. Essential oils, a new horizon in combating bacterial antibiotic resistance

Open Microbiology Journal Open Access Volume 8, Issue 1, 2014, Pages 6-14

  1. . In vitro efficacy of the essential oil from Leptospermum

scoparium (manuka) on antimicrobial susceptibility and

biofilm formation in Staphylococcus pseudintermedius

isolates from dogs

Vet Dermatol 2013; 24: 404–e87

  1. Thai Journal of Veterinary MedicineVolume 47, Issue 4, 1 December 2017, Pages 513-522 Effect of a mixture of essential oils and a plant-based extract for the management of localized superficial pyoderma in dogs: An open-label clinical trial
  2. Veterinary Dermatology Volume 23, Issue 6, December 2012, Pages 493-e95

In vitro evaluation of topical biocide and antimicrobial susceptibility of Staphylococcus pseudintermedius from dogs

  1. Acta Veterinaria-Beograd 2018, 68 (1), 95-107 Essential oils as potential anti-staphylococcal agents

  1. In vitro antibacterial activity of the manuka essential oil from Leptospermum scoparium combined with Tris‐EDTA against Gram‐negative bacterial isolates from dogs with otitis externa Song, Soon‐Young. Veterinary Dermatology Volume: 31 Issue 2 (2020)

Author Response

Dear reviewer,

Thank you for all the suggestions that you reported. We have highly appreciated your work, you’ll find a point-by-point answer to each of your questions.

Point 1: First of all the form of presentation of antimicrobial data results, i.e. MICs data and aromatograms, in Figures 2 and 1, are not readable. Could you please to summarize the MICs values and Inhibition zone diameters in form of Table simply.

Response 1: We have tried to convert both the figures into tables but they were too difficult to understand and redundant, so we decided to keep the graphical presentation and explain with more precision some peculiar aspects in captions.

Point 2: There is a lack of Conclusions, as a separate paragraph, in the manuscript. Could you complete it?

Response 2: We added this section.

Point 3: I have found in literature some essential positions (1-6 below) that are strongly connected with the studied topic, please include them into Introduction part or Discussion part.

Response 2: Thank you, we have focused on articles about the effect of natural product against S. pseudintermedius, some of your suggested references were added into Discussion.

Reviewer 2 Report

Manuscript ID: antibiotics-832456

Title: In vitro efficacy of essential oils from Melaleuca alternifolia and Rosmarinus officinalis, Manuka honey-based gel and propolis as antibacterial agents against canine Staphylococcus pseudintermedius strains

The authors investigate the antimicrobial action of two essential oils and two bee products against Staphylococcus pseudintermedius. The manuscript is interesting but needs additional revision.

Methodology:  Please provide a clearer description of how many replications of each experiment have occurred. My impression is that only one plate per isolates was completed for each of the assays with two duplicate tests? This would mean that there was replication per plate but was there replication over time? It is unclear but important to be clear.

Overall format: The manuscript has primary and secondary outcomes but the secondary outcome come first in the results and discussion. If the primary outcome is to evaluate the antibacterial effects of the natural products then these results/discussions should come first.

Keywords: Depending on the journal, words within the manuscript title should not be added to the key words. Please double check as four of the key words appears in the title.

Acronyms: Several acronyms (ST and MLST) are defined in the methods, which come at the end of the manuscript. Acronyms need to be defined at the first usage. Please check all acronyms throughout the manuscript.    

In-text citations: Most in-text citations are as follows: 

....against fungi and yeast (48-50).

....than those reported by (47), who used

However, there are citations in another format:

Line 36: Van Hoovels (2006) described.....        

Line 312: literature, Oliveira (2018)........

Line 191: in the study by Paul (11).

Line 210: S. aureus, Almasaudi (2017) has.....   

The in-text citations need to be consistent thoughout.

Figures: All figures need further description. What do the boxes and bars represent? Min-Max? 95% confidence interval? Is the line average? what does the plus mark represent? Also, some charts have tick marks for the X axis and others do not. The tick marks are a distracting particularly in figure 1A where the error (or min) bar goes to zero, appears to extend beyond axis.

Spacing of units: The spacing of units needs to be consistent throughout. For example:

Line 288: aerobically at 37°C for 2-4h

Line 315: done aerobically at 37°C for 24 h.

Line 138: value of 7.3± %

Line 141: values (0.22±0.1%)

Additional particular line items:

Line 21: a comma may need to be added after (MDR)

Line 55: of EOs and found out that.......

Please consider removing the word out. It is not needed and sounds awkward.

Line 64: anti-inflammatory, and antimicrobial abilities, as well..

Consider changing the word abilities to properties.

Line 71-72: have found out that .......

Please consider removing the word out. It is not needed and sounds awkward.

Lines 73, 211: consider reversing the order of both the 20% and 10%... as well as the citations.

It appears awkward to have the higher percentage before the lower.

Line 84: Antibiotic resistance screening showed a multifaceted panorama, Forthermore,.......

The first sentence does not have a concrete meaning; consider removing along with the Forthermore. Just state results in the results section.

Line 88: (60.9%) resulted resistant to more

Consider changing resulted to were

Line 89: More in detail, 11/14....

Consider changing more in detail to In particular

Line 92: consider removing in detail

Lines 97-98: The MLST was conducted in all the strains showing 8 different STs:........

This should be in the methods, consider revising.

Line 138: any difference between all the strains (figure 2A) TTEO had a mean MIC.....

Consider removing the word the and add a period after 2A).

Line142: statistical difference....

Consider revising difference to significance. There is more than this particular location in the paper so check other sentences as well.

Lines 153-155: As written, this is more of a method sentence. Please consider making a solid statement about the results of Figure 3

In addition, at this location Figure is capitalized but not in other locations. Please review instructions to author for correct usage.

Line 241: were picked up, and DNA was extracted using the boiling method described in the literature (53).

Consider revising sentence to more formal wording: were removed, and DNA was extracted using the boiling method (53).

Line 264: literature(59,60).

Consider adding a space between literature (59,60).

Line 312: Wordy....

Consider removing the entire first part of the sentence: The MIC was done according to the CLSI guidelines (41).

Line 322: literature.(63-67).

Consider changing the period between literature.(63-67) with a space

Line 325: No need to state that the data was stored in excel. Please consider removing the first sentence.  

Author Response

Response to Reviewer 2 Comments

Dear reviewer,

Thank you for all the suggestions that you reported. We have highly appreciated your work, you’ll find a point-by-point answer to each of your questions.

Point 1: Methodology: Please provide a clearer description of how many replications of each experiment have occurred. My impression is that only one plate per isolates was completed for each of the assays with two duplicate tests? This would mean that there was replication per plate but was there replication over time? It is unclear but important to be clear. 

Response 1: For the aromatogram we used two independent agar plates in which were tested 4 disks for each of the two EOs. The MIC were carried out in duplicate for each natural product tested. However, a more precise description was added in Materials and Methods (Lines 274-276 and Line 287).

Point 2: Overall format: The manuscript has primary and secondary outcomes but the secondary outcome come first in the results and discussion. If the primary outcome is to evaluate the antibacterial effects of the natural products then these results/discussions should come first.

Response 2: You are right. We have changed the order of result presentation following the order of outcome stated in the Introduction.

Point 3: Keywords: Depending on the journal, words within the manuscript title should not be added to the key words. Please double check as four of the key words appears in the title.

Response 3: We’ve deleted these keywords.

Point 4: Acronyms: Several acronyms (ST and MLST) are defined in the methods, which come at the end of the manuscript. Acronyms need to be defined at the first usage. Please check all acronyms throughout the manuscript.

Response 4: We’ve moved the acronyms near the first relative explanation.

Point 5: In-text citations: Most in-text citations are as follows […].

Response 5: We’ve standardized the citations.

Point 6: All figures need further description. What do the boxes and bars represent? Min-Max? 95% confidence interval? Is the line average? what does the plus mark represent? Also, some charts have tick marks for the X axis and others do not. The tick marks are a distracting particularly in figure 1A where the error (or min) bar goes to zero, appears to extend beyond axis.

Response 6: All the figures were represented as Box and Whiskers, with Min to Max. The line represents median and the + is the mean. We standardized all the captions and made consistent changes to graphics.

Point 7: Spacing of units: The spacing of units needs to be consistent throughout

Response 7: We standardized them.

Point 8: Additional particular line items […].

Response 8: Thank you for all these precious suggestions, we accepted and updated all of them.

Round 2

Reviewer 2 Report

The manuscript had improved and is more clear, thank you. Please review the manuscript one more time for the same consistency comments (see below).

Line 187: ..similar to those reported by Wegener (2018)

Please consider revising to similar to those reported (17)

Line 249:...out as described by Solyman (2013) (57).

Please consider revising to as descrobed by (57).

Author Response

Dear reviewer,

Thank you for your precious suggestions and the extreme precision in your work. 

We have modified both the citations as your suggestion.